# Empirical Distribution of the U.S. Housing Market during the Great Recession: Nonlinear Scaling Behavior after a Major Crash

**Fotios M. Siokis**

School of Economics and Regional Studies, University of Macedonia, Egnatia 156, 546 36 Thessaloniki, Greece; fsiokis@uom.edu.gr; Tel.: +30-2310-891459

**Abstract:** This study focuses on the real estate bubble burst in the US housing market during 2007–2008. We analyze the dynamics of the housing market crash and the after-crash sequence during the Great Recession. When a complex system deviates away from its typical path by the occurrence of an extreme event, its behavior is strongly characterized as nonstationary with higher volatility. With the utilization of a robust method, we present the characteristics of the aftershock period and provide useful information about the spatial distribution and the decay process of the aftershock sequence in terms of time. The returns of the housing price indices are well approximated by the empirics of a power law. Although we deal with low-frequency data, a time power-law relaxation pattern is identified. Our findings align with those in geophysics, indicating that the value of the relaxation parameter typically hovers around one and varies across different thresholds.

**Keywords:** financial distress; housing markets; housing market analysis; power law

## 1. Introduction

The major element of the 2007–2008 financial crisis—called the Great Recession—was the burst of the housing price bubble. The price of housing that started escalating from mid 90's and continued its upward path when the stock prices collapsed in 2002 during the dotcom crisis, and the economy went into a short-lived recession—came to an end at the dawn of 2008[1] with significant value losses in the property market. The burst of the housing bubble had immediate and catastrophic consequences in the banking sector, with the collapse of a number of financial institutions that, in return, triggered a major fall in equity and commodity prices.

The period from 1995 to 2007 for the housing market could be considered unique, as the prices across the nation escalated during the whole period that initially started from the West Coast and spread to the East Coast property markets. The causes of this unprecedented housing boom that lasted for more than 10 years do not coincide well with the economic fundamentals (Shiller 2008). A housing bubble that grew at the outset of the stock/dot-com bubble in early 1995 eventually led to a greater demand for housing as people accumulated substantial wealth from the run-up in stock prices, especially the prices of the internet and tech companies. It is worth noting that between 1995 and 2000, the Nasdaq exchange surged by 400% before the dot-com crash. When house prices began to rise, people started paying more for better housing, building expectations that the upward trend in prices would continue (Case and Shiller 2003)[2]. These self-fulfilling expectations were reinforced later by a historically low-interest rate environment in order to battle the 2000 recession, which further fueled the demand for housing and, consequently, the increase in prices. Hence, the main view calls for widespread speculative thinking from the part of the investors who consider that an investment in the housing market is of great importance, and as a result, boom psychology helped the spread of such thinking (Shiller 2007, 2008).

For the period of rapidly rising prices and the period of the burst, an excessive time series volatility was recorded, exhibiting a volatile (non-stationary) pattern (Figure 1). In a

complex system like the housing market, it is reasonable to question the dynamics when an extreme event causes the market to deviate significantly from its usual trajectory. In this respect, complex systems and markets in distress are modeled as self-organizing processes (Focardi et al. 2002) with heavy tales (Cont and Bouchaud 2000; Stauffer and Sornette 1999) and in a dynamical out-of-equilibrium system (Johansen and Sornette 1998; Levy et al. 1995).

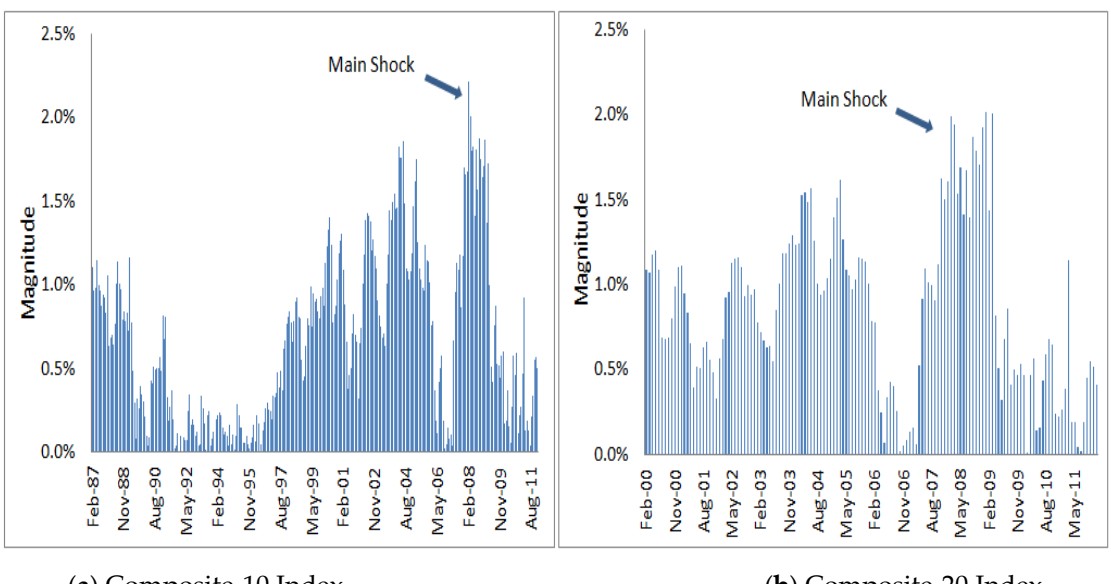

(**a**) Composite-10 Index        (**b**) Composite-20 Index

**Figure 1.** The monthly percentage change of the indexes in absolute format.

In this study, we employ an innovative approach to simulate the dynamics of housing prices, particularly in times of extreme distress. We study the behavior of selected house price indexes and their statistical characteristics in the immediate aftermath of a major price shock. Our main goal is to contribute to the ongoing literature to try to gain a better understanding of the underlying complexities of the housing market and assist regulators and investors in making more informed decisions.

The remainder of this paper is organized as follows. Section 2 reviews the literature on power law, particularly for the financial markets. Section 3 presents the data and the basic descriptive statistics. Section 4 describes the modeling of the Omori Law framework and presents the empirical properties. Section 5 discusses the results, and in Section 6, we present the concluding remarks.

## 2. Literature Review

Lillo and Mantegna (2003) studied the aftershock period of three U.S. stock market crashes by using very high-frequency data, 1-min data for a period of 60 trading days. They've concluded that the index return exhibits a non-linear behavior and is well described by a power law. In a similar vein, Weber et al. (2007) examined Power-law relaxation for a number of aftershocks following several other intermediate-size crashes. They showed that the Omori processes, which describe the decay in the rate of aftershocks of a given size with time t by a power law with an exponent close to 1, exist not only for aftermarket crashes and on very large scales but for smaller shocks. Furthermore, they demonstrated that similar Omori processes can happen in the same time span at different scales, which causes the volatility time series to exhibit self-similar characteristics, which means that some of a major crash's aftershocks can be viewed as smaller-scale sub-crashes that start Omori processes. Mu and Zhou (2008) analyzed the Chinese stock market based on 1-min (and daily) data. They found that the Omori law holds, and the scaling value increases along with the threshold value.

Petersen et al. (2010) examined the dynamics of about 219 market mainshocks. The employment of three different power laws, the Omori law, the productivity law, and the Bath law, quantified both foreshocks and aftershocks of the decay process. Siokis (2012), based on the Asian flu crisis, analyzed the crash of the Thai Baht in 1997 and its effect on the other neighborhood currencies. They concluded that there is a different crash magnitude–frequency distribution between countries. Filimonov and Sornette (2013) utilized the log-periodic power law and empirically tested it on the Shanghai Composite Index (SSE) from January 2007 to March 2008. Ohnishi et al. (2012), using data from Japan and the U.S., examined the developments in the dispersion in real estate prices and found that the land price distribution in Tokyo had a power-law tail in the late 1980s and in the U.S. data, the tail of the house price distribution tend to be heavier in those states which experience a housing bubble.

On a state level, Blackwell (2018) explored the distribution of real estate prices in Charleston County, South Carolina. He found that the best fit lies somewhere between the lognormal and power law distributions and that there is a relationship between the shape of the power law distribution and the bursting of the real estate bubble in 2007. In a recent research study by Pagnottoni et al. (2021), they studied the influence of political and socioeconomic news on several financial systems during the COVID-19 crisis. They concluded that the behavior of the foreshock and aftershock in financial markets varies, depending on the news of a pandemic. Furthermore, it was discovered that the aftershock relaxation process moves more quickly than the foreshock one. Lastly, in addition to the Omori law, Rai et al. (2022) tested the Gutenberg-Richter law for the 1987, 2008, and COVID-19 crises, using a variety of stock exchange indices and company stocks. When comparing the COVID-19 pandemic to the 2008 financial crisis, researchers have determined that the former's impact will be shorter-lived than the latter's and that the number of aftershocks follows a generalized Pareto distribution rather than Omori's power law.

We investigate the crash and the characteristics of the U.S. housing market immediately after the occurrence of the crash during the Great Recession. To our knowledge, this is the first attempt to apply such a technique and test the Omori law, a power law relaxation method, in the housing markets and by using low-frequency data, like monthly data. Up to this point, attempts were made for stock prices, internet traffic, and, of course, seismology, among others, and by using very high-frequency data starting from 1-min to daily data.

We identify the mainshock as the biggest drop in terms of the monthly percentage change of the index during the sample period, and we model the dynamics of that index returns right after the mainshock. By examining monthly percentage changes, we present an empirical regularity with respect to the number of times the absolute value of the price changes exceeds a given threshold value. The monthly changes could be considered as the sequence of the aftershocks right after the occurrence of the mainshock. In doing so, we attempt to illustrate if a time power-law relaxation is in order when the system diverges from its typical path. Testing if power law decay applies to this low-frequency data could be crucial for predicting financial crashes.

We utilize the power-law methodology because, according to Selcuk (2004), the empirical scaling laws are useful in (a) stimulating the search for interpretive frameworks, (b) imposing discipline on theory formation, and (c) unveiling properties of the space of possible underlying data generating process. Our findings and the estimation of the relaxation parameter for the housing market confirm that the Omori law accurately demonstrates the temporal variation of aftershock activity and shares similar perturbation response dynamics with earthquake studies. The concept of Omori Law offers a significant framework to model extreme events and uncertainty. During the 2008 Great Recession, the Composite-20 index exhibits greater volatility compared to the composite-10 index, with higher relaxation values at all threshold levels, meaning that larger aftershocks decay more rapidly.

### 3. Data and Descriptive Statistics

We employ the Composite-10 and Composite-20 indices along with the individual indices for the metropolitan areas of Los Angeles, San Francisco, Las Vegas, and Miami of the S&P/Case-Shiller home price indices. The Composite 10 index is a ten-city average measuring the aggregate market for single-family homes. The Composite 10 index represents a ten-city average that assesses the overall market for single-family homes, encompassing major metropolitan areas such as Boston, Chicago, Denver, Las Vegas, Los Angeles, Miami, New York, San Diego, San Francisco, and Washington Metropolitan area. The Composite 20 includes the same ten areas along with Atlanta, Charlotte, Cleveland, Dallas, Detroit, Minneapolis, Phoenix, Portland, Seattle, and Tampa.

The monthly data spans from January 1987 until December 2010, except for Composite-20, for which data are available only from January 2000. The difference in the number of data points, for the Composite 20 does not essentially affect our results, since we use the whole sample only for establishing the threshold levels as we explain below. The investigated variable is the monthly logarithmic changes of the S&P/Case-Shiller financial indices $r_{i,t}$ given by the following equation.

$$r_{i,t} = \ln(p_{i,t}/p_{i,t-1}) = \ln p_{i,t} - \ln p_{i,t-1} \tag{1}$$

where $p$ is the index and $r$ is the rate of return.

We begin our analysis by first depicting in Table 1 the summary statistics of the monthly return of the indices. In terms of the mean, the statistics reveal that almost all indices are exhibiting very similar average returns, with the markets of the State of California, particularly Los Angeles and San Francisco, having the highest at 0.4%. However, the higher average return is associated with higher volatility, as shown by the standard deviation. Also, in all indices, the values of skewness are negative, indicating that the tail of the left side of the probability density functions is longer than the right side.

**Table 1.** Descriptive Statistics of the monthly returns for all indexes.

| Index | n | Mean | std | Kurtosis | Skewness | min | max |
|---|---|---|---|---|---|---|---|
| Composite-10 | 287 | 0.3% | 0.8% | 0.59 | −0.75 | −2.8% | 2.3% |
| Composite-20 | 131 | 0.2% | 1.0% | −0.44 | −0.78 | −2.0% | 1.6% |
| Los Angeles | 287 | 0.4% | 1.2% | 0.71 | −0.53 | −3.7% | 3.4% |
| San Francisco | 287 | 0.4% | 1.3% | 1.95 | −0.87 | −4.2% | 3.2% |
| Las Vegas | 287 | 0.3% | 1.1% | 2.56 | −1.15 | −5.1% | 6.0% |
| Miami | 287 | 0.3% | 0.7% | −0.05 | −0.27 | −4.5% | 2.8% |

In Table 2, we present the date of the mainshock occurrence, the magnitude of the mainshock, and the relative shock (the magnitude of the shock divided by the relative standard deviation). Most of the main shocks occurred in the month of February 2008, with the shock in the Las Vegas market being the largest.

**Table 2.** The magnitude of the main shock and relative shock.

| Index | Date | Magnitude of Shock | Relative Shock |
|---|---|---|---|
| Composite-10 | 8 February | −2.2% | −2.80 |
| Composite-20 | 8 February | −2.0% | −2.00 |
| Los Angeles | 8 February | −3.8% | −3.20 |
| San Francisco | 8 February | −4.5% | −3.50 |
| Las Vegas | 8 January | −4.6% | −3.40 |
| Miami | 8 March | −3.9% | −3.40 |

Next, in an attempt to test scaling law behavior, we plot the log of the mean absolute return, $E|r_t|$, as a function of the log of the aggregation period. The aggregation periods

are defined as 1–10, 20, 30 and 34 months. As seen in Figure 2, all points lie on a straight line, confirming that the empirical scaling law in all indices is a power law. It seems very encouraging that the power law holds for 1 to 34 months, a very long period which corresponds to almost a 3-year period.

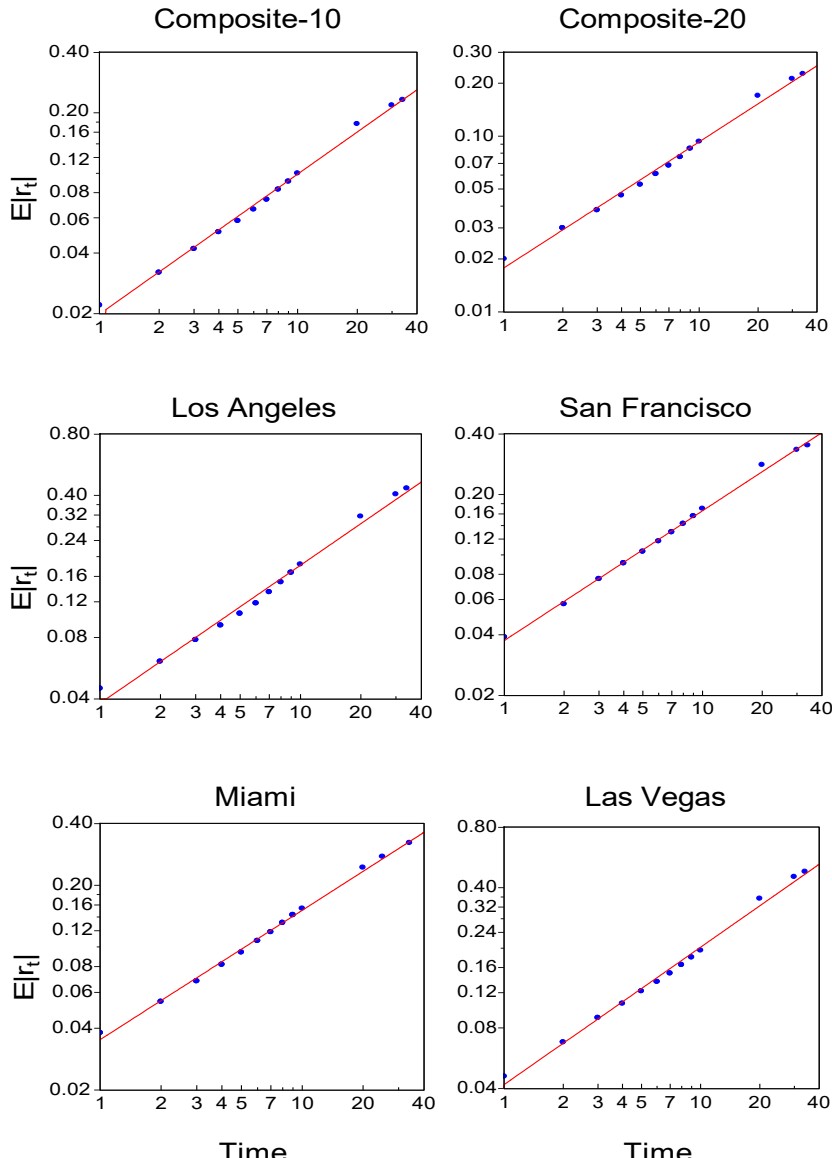

**Figure 2.** Scaling laws in different indices. The mean of the absolute return of each index $E|r_t|$ is plotted as a function of the aggregation time period on a logarithmic scale. The aggregation periods are 1–10, 20, 30, and the last data point observation (34 or 35). The mean of the absolute return of each index is plotted as a function of the aggregation time period on a logarithmic scale. The aggregation periods are 1–10, 20, 30, and the last data point observation (34 or 35).

Next, in an attempt to test scaling law behavior, we plot the log of the mean absolute return, $E|r_t|$, as a function of the log of the aggregation period. The aggregation periods are defined as 1–10, 20, 30 and 34 months. As seen in Figure 2, all points lie on a straight line, confirming that the empirical scaling law in all indices is a power law. It seems very encouraging that the power law holds for 1 to 34 months, a very long period which corresponds to almost a 3-year period[3].

### 4. Omori Law and Empirical Properties of the Aftershock Sequences

In this section, we explain the statistical robust method that we employ in order to investigate the number of times the percentage change of the index is greater than a certain threshold value based on the calculated value of the standard deviation. In other words, we examine the rate of occurrence of events as a function of time just after the eruption of the main event (mainshock). The method is based on one of the most robust and empirical relations in the geophysics and seismology discovered by Omori (Omori 1894), which states that the number n(t) per unit time of aftershock earthquakes above a given threshold measured at time t after the main earthquake decreases hyperbolically with time, like a power law, i.e., $n(t) \propto t^{-p}$.

Therefore, the Omori law is given as

$$n(t) = t^{-p} \tag{2}$$

And Utsu (1961) proposed the following modified relation in order to avoid divergence at $t = 0$, as

$$n(t) = K(t + \tau)^{-p}, \tag{3}$$

where $K > 0$, $\tau > 0$. $K$ and $\tau$ are constants where $K$ depends on the total number of events in the sequence and $\tau$ on the rate of activity in the earliest part of the sequences. $p$ is the decay parameter and varies in value, taking in empirical seismology studies from 0.5 to around 1.5 in value. By integrating Equation (3) between 0 and $t$, the cumulative number of aftershocks between the main shock and the time $t$ can be described as

$$N(t) = K[(t + \tau)^{1-p} - \tau^{1-p}]/(1 - p), \tag{4}$$

where $p \neq 1$ and $N(t) = K \ln(t/1 + \tau)$ for $p = 1$.

We trace the aftershock period for 34 months after the mainshock. We define the threshold value, i.e., the standard deviation for each series, with three different values, i.e., ($\lambda$) where $\lambda = 0.5\sigma$, $0.8\sigma$ and $1.0\sigma$, respectively, and we only use price changes, in absolute value, that exceed the threshold price. The number of these cumulative aftershocks N(t) for t = 34 is calculated, and we estimate the $p$ value from Equation (3). In all cases corresponding to different $\lambda$ threshold values, we detect a nonlinear behavior. The aftershock sequence is well described by the power law and shows that Omori Law holds after crashes of large magnitude in the real estate market. Table 3 exhibits the spatial variation in p value ranging from 0.34 to 1.71. It is interesting to point out that in all cases when the threshold value $\lambda$ increases, the relaxation exponent p increases in value as well. This means that the aftershock sequence with a greater scaling exponent decays faster. In particular, when $\lambda = 0.5\sigma$, the p value ranges from 0.32 to 0.94; when $\lambda = 0.8\sigma$, p takes values between 0.71 to 1.18 and, lastly, when $\lambda = 1.0\sigma$, the p value is within 0.89 and 1.71. Therefore, based on the above values, one could argue that the estimated exponent p differs from threshold to threshold and from index to index.

**Table 3.** Power Law and exponent *p*-value with a threshold value, λ set to 0.5σ, 0.8σ and 1.0σ.

| Index | 0.5σ | 0.8σ | 1.0σ |
|---|---|---|---|
| Composite-10 | 0.60 | 1.09 | 1.09 |
| Composite-20 | 0.70 | 1.15 | 1.71 |
| Los Angeles | 0.62 | 0.92 | 1.20 |
| San Francisco | 0.32 | 0.71 | 0.89 |
| Las Vegas | 0.84 | 0.99 | 0.99 |
| Florida | 0.94 | 1.18 | 1.39 |

In support of our findings, Figure 3 plots the cumulative number of aftershocks, N(t), and the best fit of Equation (3) for $\lambda = 0.5\sigma$, $0.8\sigma$, and $1.0\sigma$[4]. Comparison of the decay of aftershock sequences of the two composite indices reveals the significant difference in esti-

mated *p*-values among the distributions. In all aftershock sequences with various threshold levels, the estimated *p*-value for the composite 10 index is lower than the estimated *p*-value of the Composite-20 index, indicating that the activity shows a slower decay rate than the Composite-20 index. In other words, the adjustment period, after the occurrence of the mainshock, lasts longer in the case of the Composite-10 index, which comprises the top 10 U.S. metropolitan areas. The higher *p*-value for the Composite-20 Index is associated with higher variability.

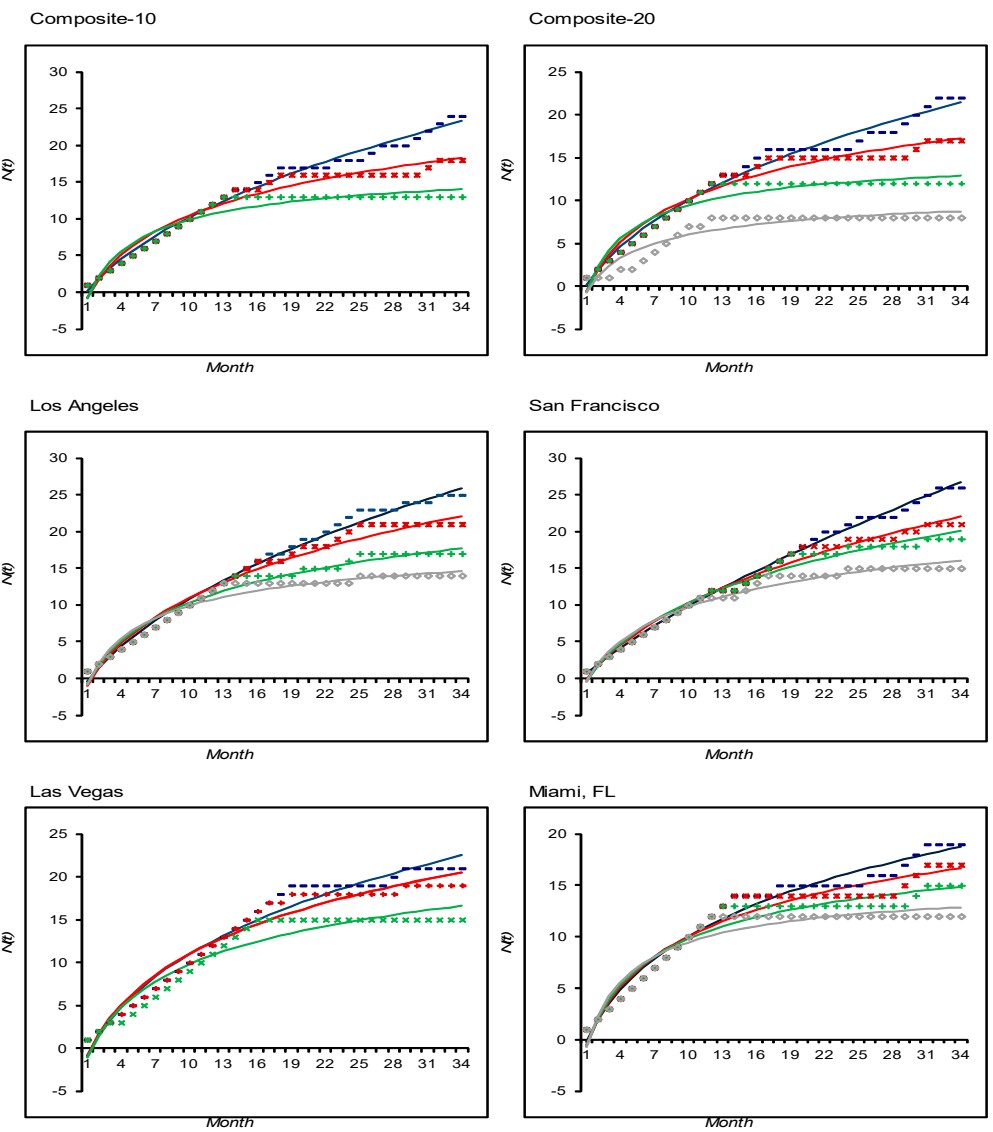

**Figure 3.** Cumulative number $N(t)$ of aftershocks exceeding the threshold λ. Solid lines are the best fit for Equation (4). The blue line depicts the curves with values exceeding λ = 0.5σ, the red line with λ = 0.8σ, the green line with λ =1.0σ (and the brown line with λ = 1.5σ in some cases where there are enough points higher than the value of 1.5σ). The period is for 34 months, right after the main shock.

Further research on selected individual metropolitan areas reveals that the higher *p*-value is observed in the Miami index regardless of the threshold level, meaning that larger aftershocks decay faster, or that in the case of Miami there is a rapid relaxation and quicker adjustment towards to a new equilibrium.

Further research on selected individual metropolitan areas reveals that the higher *p* value is observed in the Miami index regardless of the threshold level, meaning that larger aftershocks decay faster or that, in the case of Miami, there is a rapid relaxation and quicker adjustment toward a new equilibrium.

## 5. Discussion

The statistical properties of the occurrence of aftershocks recently have taken center stage in connection with the processes of financial bubble generation. The characteristics of aftershock sequences yield useful information about the spatial distribution, the total number of aftershocks, and the decay rate of the sequence with time. Also, aftershock sequences offer source properties of large mainshocks/crashes because a very large number of events occur over a short period of time. In our case, the aftershock sequence of the Great Recession housing market crash reveals that all cases investigated are well approximated by the power law, like the other financial markets, and display a remarkable power law relaxation for a long period of time. Particularly, after a major drop in the price housing indices, the price volatility above a certain threshold shows power-law decay, which is in line with the well-known Omori law in geophysics. Even though we use low-frequency data, our findings demonstrate that the housing market shares with earthquakes and with other financial market crashes, like stock exchange markets, a common scale-invariant feature in the temporal patterns of aftershocks. Hence, housing markets can also be considered a complex system with non-stationary temporal behavior, where Omori law seems to be universal.

The paper's results are in line with the results of other studies that examined the Omori law dynamics in stock market behavior, such as Lillo and Mantegna (2003), Weber et al. (2007), and Petersen et al. (2010), even though our data is based on a lower frequency. We deviate from the results of Mu and Zhou (2008), particularly with the size of the parameter p, probably because of the government intervention in the Chinese stock market.

## 6. Conclusions

The principal conclusion of the paper is that aftershock statistics do vary between sectors and time periods. Omori Law provides an important foundation for modeling severe events and uncertainty. The background turbulence of the Composite-20 aftershock sequence may cause a higher *p*-value in comparison with the Composite-10 events. Because of the estimated higher *p*-value, the activity shows a relatively faster decay rate in the Composite-20 index. The higher value indicates that larger aftershocks decay more rapidly, and the turbulence period for the Composite-20 index is shorter. In other words, the adjustment period towards a new equilibrium lasts less in the case of Composite-20 compared with the Composite-10 index, which comprises the big 10 U.S. metropolitan areas. This result is reinforced by the comparison of the individual metropolitan areas, like Los Angeles, San Francisco, Las Vegas, and Miami, with their estimated *p*-values in almost all cases to be lower relative to the composite-20 *p*-value. Hence, there is a significant variation in the decay behavior for different time periods and indices. The price indices exhibit self-similar features, indicating that the Omori processes hold both high and smaller-scale magnitude shocks. In conclusion, the correlations between housing market prices are essential, especially for risk estimation.

**Funding:** This research received no external funding.

**Data Availability Statement:** Data available on request.

**Conflicts of Interest:** The author declares no conflict of interest.

## Notes

[1]   Prices had started going down since late 2006, but according to Case/Shiller index the major drop in the housing prices was recorded in February 2008 when composite-10 index decreased by 2.2% month on month.

[2]   According to Case and Shiller (2003), when people buying houses as an investment is one of the characteristics of the housing bubble, meaning that people form expectations for future home appreciation.

[3]   In some cases there is a small deviation from linearity but this is due to monthly seasonality in absolute returns of the indices.

[4]   Figure 3 includes, in some cases, when there is available data, the best fit for the cumulative number of aftershocks with $\lambda = 1.5\sigma$ in order to show that power law relaxation still holds.

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
