# Peer review of "Empirical Distribution of the U.S. Housing Market during the Great Recession: Nonlinear Scaling Behavior after a Major Crash"

_jrfm, doi:10.3390/jrfm17030130_

Round 1

Reviewer 1 Report

Comments and Suggestions for Authors

Comments for the authors

1.       This paper is about scaling law (scaling behavior and power law) but the author fails to provide its clear definition and purposes.   There is not much motivation for the reasons behind your research.  I suggest that you add more details on the “what” and the “why” in your Introduction section.  This is an easy fix that can greatly improve interest to readers.

2.       At the end of your Introduction, you have a one sentence paragraph.  There, you can add a little summary of your objectives and your findings.

3.       In Section 2, you listed the cities in the Composite 10 index.  You only listed only nine instead of ten.

4.       Your data span from 1987 (or 2000) to 2010.  It’s 2024 now. There is more than 10 years of missing data.  No biggie.  I know that it is hard to add more data sometimes.

5.       In the Descriptive Statistics table, please include columns for the Min and Max.  Since you look at absolute values, I would like to see if they cross zero.

6.       You forgot the “s” in Las Vegas.

7.       There are inconsistencies in your results.   For 0.5, you claimed p is between 0.34 to 0.94.  According to table 3, it is between 0.32 and 0.94.   Which is the correct one?  Same goes for 1.0, between 0.99 and 1.71 or 0.89 and 1.71?

8.       Most of your references are more than a decade old, why don’t you include some of the more recent findings?

Comments on the Quality of English Language

You only need to make a few small fixes.  

Author Response

1st Reviewer

Comments for the authors

  1. This paper is about scaling law (scaling behavior and power law) but the author fails to provide its clear definition and purposes.   There is not much motivation for the reasons behind your research.  I suggest that you add more details on the “what” and the “why” in your Introduction section.  This is an easy fix that can greatly improve interest to readers.

 Ans. I reworked some of the paragraphs for the Introductions and include more information.

  1. At the end of your Introduction, you have a one sentence paragraph.  There, you can add a little summary of your objectives and your findings.

Ans. I’ ve rearranged and included parts of the introduction and separated the literature review.

       “ In this study, we employ an innovative approach to simulate the dynamics of housing prices, particularly in times of extreme hardship. We study the behavior of selected house price indexes and their statistical characteristics in the immediate aftermath of a major price shock. Our main goal is to contribute to the ongoing literature, and try to gain a better understanding of the underlying complexities of the housing market, and assist the regulators and investors to make more informed decisions.

“The remainder of this paper is organized as follows. Section 2 reviews the literature of the power law particularly for the financial markets. Section 3 presents the data and the basic descriptive statistics. Section 4 describes the modeling of the Omori Law framework and presents the empirical properties. Section 5 discusses the results and in section 5 we present the concluding remarks”.

  1. In Section 2, you listed the cities in the Composite 10 index.  You only listed only nine instead of ten.

Ans. Done

  1. Your data span from 1987 (or 2000) to 2010.  It’s 2024 now. There is more than 10 years of missing data.  No biggie.  I know that it is hard to add more data sometimes.

Ans. Yes, but the study deals only with the 2008 crisis.

  1. In the Descriptive Statistics table, please include columns for the Min and Max.  Since you look at absolute values, I would like to see if they cross zero.

Ans. Done

  1. You forgot the “s” in Las Vegas.

Ans. Okay

  1. There are inconsistencies in your results.   For 0.5, you claimed p is between 0.34 to 0.94.  According to table 3, it is between 0.32 and 0.94.   Which is the correct one?  Same goes for 1.0, between 0.99 and 1.71 or 0.89 and 1.71?

Ans. Yes spot on.  They’re corrected!

  1. Most of your references are more than a decade old, why don’t you include some of the more recent findings?

Ans. I’ve includes two recent ones in the literature sectin as well as in the bibliography.

 Pagnottoni, P., Spelta, A., Pecora, N., Flori, A., Pammolli, F. (2021). Financial earthquakes: SARS-CoV-2 news shock propagation in stock and sovereign bond markets. Physica A: Statistical Mechanics and its Applications, Vol 582, 15, 126240.

Rai, A., Mahata, A.,  Nurujjaman, M. and  Prakash, O., (2022). Statistical properties of the aftershocks of stock market crashes revisited: Analysis based on the 1987 crash, financial-crisis-2008 and COVID-19 pandemic. International Journal of Modern Physics Vol. 33, No. 02, 2250019.

 Minor points:

1- L. 11: non stationary --> non-stationary

2- L. 23: their --> its

3- L. 54: out-of equilibrium --> out-of-equilibrium

4- L. 56: The caption of figure 1 is vague and incomplete.

5- L. 90: changes we --> changes, we

6- L. 100: a) in --> in a)

7- L. 140: In figure 2, E|rt| has to be defined clearly.

8- L. 145: statistical robust --> robust statistical

9- L. 147: In other words we --> In other words, we

10- L. 157: constants where --> constants, where

11- L. 162: Equation 4 has to be rearrange as a formula. It has also missed parentheses.

12- L. 166: (λ) where --> (λ), where

13- L. 174-179: Why do the reported scaling exponents (p) in the text and in the table 3 have different values?! 

14- L.189: In figure 3, the extracted data and fitting results for different thresholds have to be illustrated in different colors and different markers. Figure should have also legend in each panel.

15- L. 189: In figure 3, why doesn’t the author fit data for λ=1.5σ in all panels?

16- L. 212: nonstationary --> non-stationary

17- L. 216-218: Why does Composite-20 has higher p value in comparison with Composite-10?

Ans. All of the above were taking into consideration and corrected them. Also, noted in the manuscript, that for the last comment, “the higher p-value  for the Composite 20 is associated with higher variability”.

Reviewer 2 Report

Comments and Suggestions for Authors

 Minor points:

1- L. 11: non stationary --> non-stationary

2- L. 23: their --> its

3- L. 54: out-of equilibrium --> out-of-equilibrium

4- L. 56: The caption of figure 1 is vague and incomplete.

5- L. 90: changes we --> changes, we

6- L. 100: a) in --> in a)

7- L. 140: In figure 2, E|rt| has to be defined clearly.

8- L. 145: statistical robust --> robust statistical

9- L. 147: In other words we --> In other words, we

10- L. 157: constants where --> constants, where

11- L. 162: Equation 4 has to be rearrange as a formula. It has also missed parentheses.

12- L. 166: (λ) where --> (λ), where

13- L. 174-179: Why do the reported scaling exponents (p) in the text and in the table 3 have different values?! 

14- L.189: In figure 3, the extracted data and fitting results for different thresholds have to be illustrated in different colors and different markers. Figure should have also legend in each panel.

15- L. 189: In figure 3, why doesn’t the author fit data for λ=1.5σ in all panels?

16- L. 212: nonstationary --> non-stationary

17- L. 216-218: Why does Composite-20 has higher p value in comparison with Composite-10?

Comments on the Quality of English Language

1- L. 11: non stationary --> non-stationary

2- L. 23: their --> its

3- L. 54: out-of equilibrium --> out-of-equilibrium

4- L. 56: The caption of figure 1 is vague and incomplete.

5- L. 90: changes we --> changes, we

6- L. 100: a) in --> in a)

7- L. 140: In figure 2, E|rt| has to be defined clearly.

8- L. 145: statistical robust --> robust statistical

9- L. 147: In other words we --> In other words, we

10- L. 157: constants where --> constants, where

11- L. 162: Equation 4 has to be rearrange as a formula. It has also missed parentheses.

12- L. 166: (λ) where --> (λ), where

13- L. 174-179: Why do the reported scaling exponents (p) in the text and in the table 3 have different values?! 

14- L.189: In figure 3, the extracted data and fitting results for different thresholds have to be illustrated in different colors and different markers. Figure should have also legend in each panel.

15- L. 189: In figure 3, why doesn’t the author fit data for λ=1.5σ in all panels?

16- L. 212: nonstationary --> non-stationary

17- L. 216-218: Why does Composite-20 has higher p value in comparison with Composite-10?

Author Response

 Minor points:

1- L. 11: non stationary --> non-stationary

2- L. 23: their --> its

3- L. 54: out-of equilibrium --> out-of-equilibrium

4- L. 56: The caption of figure 1 is vague and incomplete.

5- L. 90: changes we --> changes, we

6- L. 100: a) in --> in a)

7- L. 140: In figure 2, E|rt| has to be defined clearly.

8- L. 145: statistical robust --> robust statistical

9- L. 147: In other words we --> In other words, we

10- L. 157: constants where --> constants, where

11- L. 162: Equation 4 has to be rearrange as a formula. It has also missed parentheses.

12- L. 166: (λ) where --> (λ), where

13- L. 174-179: Why do the reported scaling exponents (p) in the text and in the table 3 have different values?! 

14- L.189: In figure 3, the extracted data and fitting results for different thresholds have to be illustrated in different colors and different markers. Figure should have also legend in each panel.

15- L. 189: In figure 3, why doesn’t the author fit data for λ=1.5σ in all panels?

16- L. 212: nonstationary --> non-stationary

17- L. 216-218: Why does Composite-20 has higher p value in comparison with Composite-10?

Ans. All of the above were taking into consideration and corrected them. Also, noted in the manuscript, that for the last comment, “the higher p-value  for the Composite 20 is associated with higher variability”

Reviewer 3 Report

Comments and Suggestions for Authors

This is a concise, compehensive and interesting paper experimentally investigating the after-shock log-return statistics of the US housing marker bubbke burst of 2008. The author establises the complex dynamical behavior of the housing market in crisis by showing a) that the mean abs. value of monthly returns after the main event follow a power law with respect to the averaging time period and b) that the frequency of large abs. value returns (greater than some particular threshold) also follow a power law as a function of time after the main "seisimc" event.

The methodology and analysis is scientifically sound and I think on the overall the paper is very interesting and deserves to be published with mostly minor modifications.

However, I still have a slightly more "major" suggestion for a modification to be considered by the author:  supply more information on the statistics of the non-linear fits to equation 3 (for main result b) as above) for the various thresholds. The best-fit values of exponents p (as is apparent from the fit curves in Fig. 3) appear to have a rather large uncertainty (this is understandable as the data is quite sparse) but the author does not report it. I think the std in the values of p it should be reported next to each best fit value in Table 3. In addition, Pearson's R of the best fits could be reportes inline the text so that the reader gets a feeling of the believability of the fits. The author should include 1-2 sentences in the analysis of figure 3 and results of Table 3 discussing the quality of the fits to equation 3.

Based on the apparent large unscertainties in p, the way to better convince the reader that the decay follows Omura law indeed is to generate a surrogate series of after shoch log-returns where returns follow a Gaussian distribution of the same std as the original data. Applying the same procedure to the surrogate series for rare (large return) events and plotting in Figure 3 will clearly show that the follow expenential decay rather than power law. Comparison between actual series and surrogate series would make the author's claim much stronger.

More minor suggestions:

I do not understand how the fact that the aftershock sequancies have power law behaviors can be useful for 'forecasting' periods of ciris in financial markets as the author claims in the conslusions and I think one more place in the paper. It seems that it takes a long period to establish this after-shock period power-law behavior when it would be too late to exploit the knowledge that we have gone through a market crash. I suggest that either the author elaborates a bit more on the forecasting ability of the results or else eliminate this claim form the paper. I think the fact that the after-shock dynamics follows power-law behaviors is already interesting enough.

Line 63: What are ‘Omori processes’ and (later on) the Omori law? Maybe a short phrase linking this to the previous sentences and also a citation here would be helpful for the non-familiar reader who has to wait until line 150 which is much later in the text.

Lines 154-156: It seems that equations 2 and 3 are exactly the same. Most probably τ does not belong in equation 2. Also, p must be strictly positive, a fact that is not mentioned.

Figure 3: The plots in this figure are a bit confusing as they display a lot of data at once. I think that the experimental data as well as the best fit lines for different thresholds λ should be indicated with different colors and maybe different point symbols so that it is clearer which fit curve belongs to which dataset.

Author Response

2nd Reviewer.

This is a concise, compehensive and interesting paper experimentally investigating the after-shock log-return statistics of the US housing marker bubbke burst of 2008. The author establises the complex dynamical behavior of the housing market in crisis by showing a) that the mean abs. value of monthly returns after the main event follow a power law with respect to the averaging time period and b) that the frequency of large abs. value returns (greater than some particular threshold) also follow a power law as a function of time after the main "seisimc" event.

The methodology and analysis is scientifically sound and I think on the overall the paper is very interesting and deserves to be published with mostly minor modifications.

Ans. Thank you

However, I still have a slightly more "major" suggestion for a modification to be considered by the author:  supply more information on the statistics of the non-linear fits to equation 3 (for main result b) as above) for the various thresholds. The best-fit values of exponents p (as is apparent from the fit curves in Fig. 3) appear to have a rather large uncertainty (this is understandable as the data is quite sparse) but the author does not report it. I think the std in the values of p it should be reported next to each best fit value in Table 3. In addition, Pearson's R of the best fits could be reportes inline the text so that the reader gets a feeling of the believability of the fits. The author should include 1-2 sentences in the analysis of figure 3 and results of Table 3 discussing the quality of the fits to equation 3.

Based on the apparent large unscertainties in p, the way to better convince the reader that the decay follows Omura law indeed is to generate a surrogate series of after shoch log-returns where returns follow a Gaussian distribution of the same std as the original data. Applying the same procedure to the surrogate series for rare (large return) events and plotting in Figure 3 will clearly show that the follow expenential decay rather than power law. Comparison between actual series and surrogate series would make the author's claim much stronger.

Ans. First I would like to thank the reviewer for the above remarks which are really significant, and the author believes that they could contribute greatly to the quality and advancement of the paper. But due to the limited time, I’m not in a position to accommodate them as of now. I will certainly take them into consideration for the ongoing research that I do, especially for the crude oil markets.

More minor suggestions:

I do not understand how the fact that the aftershock sequancies have power law behaviors can be useful for 'forecasting' periods of ciris in financial markets as the author claims in the conslusions and I think one more place in the paper. It seems that it takes a long period to establish this after-shock period power-law behavior when it would be too late to exploit the knowledge that we have gone through a market crash. I suggest that either the author elaborates a bit more on the forecasting ability of the results or else eliminate this claim form the paper. I think the fact that the after-shock dynamics follows power-law behaviors is already interesting enough.

Ans. Yes correct. It does take some time for the aftershocks  (since we deal with monthly data)  but still the results could be use to form a model for forecasting purposes. But I did take out (eliminate)  the forecasting  claim

Line 63: What are ‘Omori processes’ and (later on) the Omori law? Maybe a short phrase linking this to the previous sentences and also a citation here would be helpful for the non-familiar reader who has to wait until line 150 which is much later in the text.

Ans. I have included a definition of the process taken from the Weber et al., paper.

Lines 154-156: It seems that equations 2 and 3 are exactly the same. Most probably τ does not belong in equation 2. Also, p must be strictly positive, a fact that is not mentioned.

Ans. Yes it was the same and differ format. I meant to have to Omori law first and  then the modified (Omori and Utsu) . It is corrected.

Figure 3: The plots in this figure are a bit confusing as they display a lot of data at once. I think that the experimental data as well as the best fit lines for different thresholds λ should be indicated with different colors and maybe different point symbols so that it is clearer which fit curve belongs to which dataset.

Ans. I went ahead and change the color as well as the symbols for each fit (by the different σ)